# LEARNING EXTRAPOLATIVE SEQUENCE TRANSFORMATIONS FROM MARKOV CHAINS

## ABSTRACT

Most successful applications of deep learning involve similar training and test conditions. However, for some generative tasks, samples should improve desirable properties beyond previously known values, which requires the ability to generate novel hypotheses that *extrapolate* beyond training data. While large language models have been successfully extended to a variety of sequence modeling problems, greedy autoregressive sampling can struggle to explore the solution space sufficiently to extrapolate, especially when the properties of interest are global to the sequence. On the other hand, sequence-level sampling methods such as Markov chain Monte Carlo (MCMC) offer theoretical guarantees about capturing the distribution of interest, but suffer from the curse of dimensionality in discrete structured spaces. We propose a new approach that bridges the gap between MCMC and autoregressive sampling, which may be viewed as off-policy reinforcement learning. Our approach uses selected states from Markov chains as a source of training data for an autoregressive inference network, which is then able to generate novel sequences at test time that extrapolate along the sequence-level properties of interest. The proposed approach is validated on three problems: protein sequence design, text sentiment control, and text anonymization. We find that the learned inference network confers many of the same (and sometimes better) generalization benefits compared to the slow sampling process, but with the additional benefit of high sample efficiency.

## 1 INTRODUCTION

In creative tasks such as scientific discovery, a key requirement is the ability to *extrapolate* beyond existing knowledge. For example, automating the generation of novel hypotheses is central to mathematical discovery, biological sequence design, molecular optimization, and the creation of new materials (Romera-Paredes et al., 2024; Fu et al., 2023; Jain et al., 2022; Trabucco et al., 2022; Gao et al., 2022). Beyond scientific discovery, extrapolation is necessary in many creative applications, such as writing assistants for creative writing (Swanson et al., 2021; Gómez-Rodríguez & Williams, 2023). It is natural to wonder if large-scale generative training affords extrapolation as an emergent ability (Schaeffer et al., 2024). Unfortunately, prior work has found that state-of-the-art foundation models can struggle on tasks requiring extrapolation (Dziri et al., 2023; Chakrabarty et al., 2024). Notably, Lu et al. (2024) compare different reasoning and inference strategies, finding that the only strategy to successfully increase sample diversity is Monte Carlo search, which typically suffers from low sample efficiency and can produce degenerate samples in high-dimensions (Holtzman et al., 2019).

How can we *efficiently* extrapolate beyond the training data? We build on a recent approach which leverages the de-noising ability of masked language models (MLM) to extrapolate (Padmakumar et al., 2023). The idea is to generate many sequence transformations that improve the target objective as evaluated by a trained scorer model, and then to supply these transformations as training data for a greedy extrapolative model. To search for suitable transformations to create this augmented training set, Padmakumar et al. (2023) apply a random mask to sequences in the training data, which are then in-filled by sampling from an MLM. Samples are kept if the improvement in the objective between the sampled sequence and the original sequence is within a fixed range. This process is repeated for a fixed number of steps with the goal of identifying transformations that make incremental improvements to the objective. A key assumption is that, after training a sequence-to-sequence

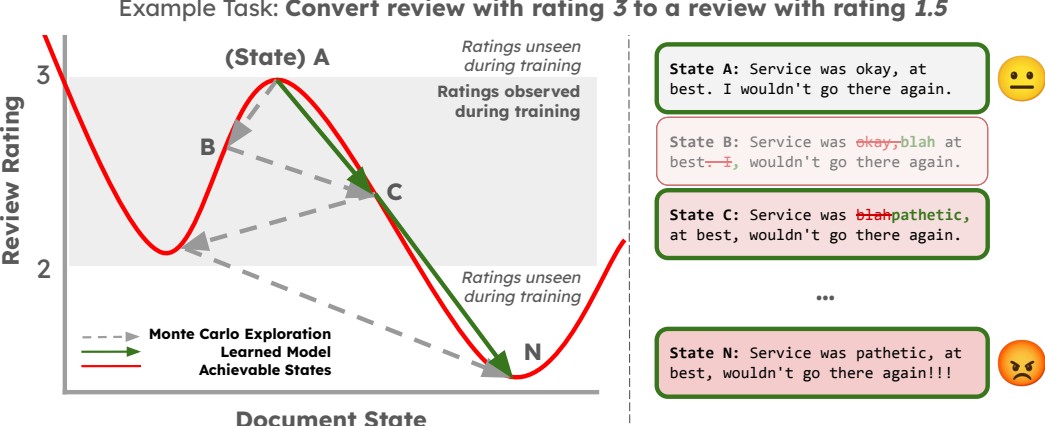

Figure 1: The sentiment extrapolation task (§4.2, Padmakumar et al. (2023)) requires generating reviews with ratings beyond the range observed at training time. Here, we illustrate the search process using a toy 1D representation of the features (x-axis) and rating (y-axis). Monte Carlo exploration can produce reviews that extrapolate, but many steps are required. However, once good state sequences have been discovered, we can sub-sample the transitions that decrease the rating (A → C → N) and use them to learn an extrapolative model. The reviews shown to the right for states B, C, and N are actual reviews generated by our method, while A is a genuine review from the validation data.

model on the selected transformations, composing more than one transformation can lead to effective extrapolation. However, although this approach was found to be successful in extrapolating beyond the training region for some tasks, its success is critically dependent on the choice of a number of sensitive hyper-parameters, including a threshold on the relative improvement from different transformations and a fixed number of iterative decoding steps.[1]

In this paper, we first seek to better understand how generative models, in particular models trained using in-filling objectives (Bavarian et al., 2022; Tay et al., 2022), implicitly capture knowledge that can be leveraged for extrapolative generation. To do so, we formalize the process of searching for sequences that score highly under the target sequence-level objective as approximate inference in an energy-based model (EBM) (LeCun et al., 2006). This model is specified via an unnormalized score (negative energy), which can incorporate multiple criteria via a product-of-experts (Hinton, 2002; Mireshghallah et al., 2022). The experts will typically include a measure of fluency or faithfulness along with a task-specific sequence-level objective for extrapolation. While exact inference in EBM is intractable, the MLM provides a convenient and effective *proposal distribution* for a Metropolis-Hastings (MH) sampler, which under mild assumptions approximates the distribution over sequences defined by the EBM (Goyal et al., 2021). Beyond providing a conceptual framework for understanding the search process, we find this formulation also provides practical benefits in terms of improved generalization and robustness (§4).

However good the proposal distribution, MH still suffers from all the aforementioned limitations. Therefore, we fine-tune a model using the Markov chains resulting from MH as training data. Our objective in doing so is to generate sequences that achieve scores in the extrapolation range in *as few steps as possible*. This is illustrated in Figure 1 for the controlled task of review generation (§4.2). Using every transition in the Markov chains is clearly undesirable, since some transitions may fail to improve the score or result in *worse* scores. As a result, we explore several strategies to sub-sample state sequences from the complete chains, including adaptive schemes based on the relative improvement in energy. While the model we fine-tune has an autoregressive parametrization (§3), by selecting a variable number of transitions from the Markov chains, we implicitly learn a non-autoregressive model that transforms an initial sequence (token-by-token) a variable number of times to improve the score beyond the training range. By further incorporating the sequence-level score at each step of generation—similar to reward-to-go in sequence modeling approaches to reinforcement

---

[1]In a personal communication, the authors report that their procedure exhibits large variance, and indeed we are unable to reproduce published results using the code released by Padmakumar et al. (2023).

learning (Janner et al., 2021)—the model can learn to incorporate this feedback, for example to help determine when to stop generating.

**Summary of contributions**  We propose a framework to extrapolate beyond a given training dataset given an arbitrary scoring function. Our approach leverages existing components, namely pre-trained language models trained using de-noising objectives, to explore the space of sequence-to-sequence transformations and their impact on the target objective. We formalize this process as MCMC, and consider a variety of strategies to select training data from the resulting Markov chains to fine-tune a model to generate novel sequences. In particular, we propose a multi-step generative process in which, starting from an initial state, the properties of interest are optimized in multiple rounds, similar to non-autoregressive generation. We evaluate our model on three tasks: protein engineering, sentiment style transfer, and anonymization.[2] In some cases, we find that our model, $q_\theta$, can achieve competitive results with MCMC and other baselines, but using a significantly smaller number of steps (§4). In other cases, specifically §4.1, we find that the fine-tuned model achieves significantly better extrapolation.

## 2  PROBLEM STATEMENT

We consider sequence-level search problems where the task is to generate novel sequences $x \in \mathcal{X}$ that satisfy one or more properties of interest $y \in \mathcal{Y}$. Given a candidate sequence $x$, we assume that an oracle $o(x) \in \mathcal{Y}$ may be consulted to assess $x$, but that it may be expensive to consult. For example, assessing a novel candidate may involve conducting physical experiments or running expensive simulations (as in the protein task described in §4.1), and therefore we wish to minimize the number of evaluations of $o(x)$ when searching for new sequences. At training time, we observe sequences $x \in \mathcal{X}^{\text{train}}$ with properties taking values in the training range $o(x) \in \mathcal{Y}^{\text{train}}$, and our objective is to fit a generative model $q_\theta$ such that samples $x' \sim q_\theta$ successfully extrapolate beyond the training data for the property of interest: $o(x') \notin \mathcal{Y}^{\text{train}}$. For example, for the sentiment task, the training range consists of ratings between 2 and 4 stars, and the extrapolation range consists of ratings that are highly negative (less than 2-stars) or highly positive (greater than 4-stars). If the oracle is expensive to consult at training time, we instead assume access to a *guide* $s(x)$ that provides a computationally tractable estimate $s(x)$ of the oracle score $o(x)$. For example, $s(x)$ may be a neural network trained to predict properties of $x$ based on a database of previous experiments with hypothesized sequences $x$ and measured outcomes $o(x)$. At test time, we generate $x' \sim q_\theta$ and then evaluate performance under the oracle $o(x')$. Overall, the central problem is how to fit $q_\theta$ without overfitting the training data and in such a manner as to enable extrapolation.

## 3  METHOD

**Extrapolative generation**  We are interested in generating novel sequences that extrapolate beyond a given training distribution for one or more attributes of interest. Since the attributes of interest may be properties of the complete sequence, we consider the family of energy-based models (EBM) (LeCun et al., 2006), where the log-probability of an event is proportional to a sequence-level score $s(x)$. Similar to rewards in reinforcement learning (RL), this parametrization affords considerable flexibility in choosing appropriate scoring functions for the task. The scoring function may be fast (for instance, a small neural classifier) or slow (such as using a slow evaluation process to calculate the folding energy of a protein). However, in either case, exact sampling from an EBM is intractable since the partition function $Z$ involves a sum over all possible sequences. In our experiments, we often include multiple terms in our energy, which are combined in a product of experts $\ln p(x) = \alpha_1 s_1(x) + \alpha_2 s_2(x) + \ldots - \ln Z$, weighted with scalar hyperparameter $\alpha$. For example, for the sentiment task, we include Hamming distance to the original review in addition to the sentiment rating.

**Masked-infilling language models**  In general, while MCMC can be used to (approximately) draw samples from EBMs, the algorithm suffers from the curse of dimensionality which can manifest as slow mixing and in failures to identify modes of the energy landscape. These issues can be mitigated

---

[2]We additionally include a minimal demonstration of the key idea on a toy task in Appendix H.

in part by choosing effective proposal distributions. Crucially for our method, language models trained with mask-infilling objectives can serve as effective proposal distributions (Goyal et al., 2021; Mireshghallah et al., 2022).[3] This fact allows us to obtain proposals using existing pre-trained language models. Specifically, we use the Metropolis-Hastings (MH) algorithm which uses a proposal distribution $q(x' \mid x)$ to draw candidate states $x'$ given the current state $x$. These proposals are then either accepted, in which case $x'$ is taken as the new state, or rejected in which case $x' = x$, according to the standard MH acceptance criterion. To implement $q$, we mask at random subset of the current state $x$, and then *infill* the masked sequence based on a self-supervised pre-training process (Devlin et al., 2019; Lewis, 2019; Raffel et al., 2023).

**Training** $q_\theta$    Although an effective proposal distribution can improve the mixing time of MCMC, the amount of iterations required to identify modes of the distribution may still be prohibitive. Our approach will therefore be to fine-tune a separate model $q_\theta$ from which it is efficient to draw samples that ideally extrapolate beyond the scores explored during MCMC (see Appendix H for a simple demonstration of this idea). This is similar to off-policy RL, where we use MCMC as a particular kind of exploration policy to generate training episodes. We imbue $q_\theta$ with specific inductive biases to encourage extrapolation beyond the training data. In particular, rather than sampling $x \sim q_\theta$ directly, we allow generation to proceed via multiple intermediate states $x_1, x_2, \ldots, x_N$. The intuition for this strategy, which is borne out in our experiments (§4), is that it is easier to learn a conditional transformation $q_\theta(x_n \mid x_1, x_2, \ldots, x_{n-1})$ than directly sample the structured objects $x$. Unlike the state-to-state transitions in MCMC however, $q_\theta$ is biased to be greedy: it aims to continually improve the energy from state-to-state and in general may avail of information from the complete history of previous states $x_1, x_2, \ldots$, *as well as associated scores* $s(x_1), s(x_2), \ldots, s(x_{n-1})$, when producing the next state $x_n$. By conditioning on the scores, the policy has the ability to incorporate these into planning, not unlike the sequence model RL formulations proposed by Janner et al. (2021); Chen et al. (2024).

**Autoregressive refinement**    To fit $q_\theta$, we assume access to a training dataset providing one or more initial states, from which we sample state trajectories using MCMC. We then create *training episodes* $(x_1, s_1), (x_2, s_2), \ldots, (x_N, s_N)$ by sub-sampling state sequences from the complete trajectories. We discuss several strategies for this in §3.1. The training episodes are encoded as a sequence of tokens:

$$x_0 \; \texttt{<seq0>} \; x_1 \; \texttt{<seq1>} \; s_1 \; x_2 \; \texttt{<seq2>} \; s_2 \; \ldots \; x_n \; s_n \; \texttt{<stop>}$$

Above, $\texttt{<seq}i\texttt{>}$ and $\texttt{<stop>}$ are distinguished symbols encoded either as special vocabulary terms or as strings in a pre-trained model, $s_i$ are scalar scores, and $x_i$ are token sequences of possibly variable length. Then $q_\theta$ is trained using teacher forcing to generate each token of each intermediate state $x_i$ (for $i > 0$) conditioned on all previous states $x_0, x_1, \ldots, x_{i-1}$. As previously mentioned, the concrete advantage to formulating inference in this way is that revisions can condition on previously generated sequences and energy scores. As an ablation, we also experiment with a Markov variation that only conditions on the previous state, which performs well in certain settings.

**Inference**    Since $q_\theta$ has a simple autoregressive structure, generating from the model can be done in a variety of ways, including forward sampling and beam search. We note that in principle constrained decoding techniques could be used to enforce adherence to the structure above, but we did not find this necessary in practice. After generating each intermediate state $x_i$, the sequence is either scored using $s(x_i)$ and the result deterministically appended to the sequence, or $q_\theta$ learns to *predicts* the sequence score.[4] When $\texttt{<stop>}$ is generated from the model, the final state $x_n$ is taken to be the sample.

### 3.1    CREATING TRAINING EPISODES

Creating training episodes consisting of the *entire* Markov chain, which could include hundreds or thousands of states, is undesirable. Ideally, $q_\theta$ should be computationally efficient at inference time, only generating a small number of intermediate states before producing the $\texttt{<stop>}$ symbol. As a

---

[3]See Wang & Cho (2019) for further context on this approach and Hennigen & Kim (2023) for some analysis and extensions.

[4]Another possibility is to consult the oracle at *intermediate* states of generation, although we do not directly evaluate this version in our experiments.

result, we require relatively short training episodes. Note also that the sampling method might explore high-energy regions of the state space, and it may be sub-optimal to include such exploration in the training episodes; therefore, we ideally want to select state transitions from the complete sample that result in a decreased energy. We examine several strategies for selecting states.

**Uniform thinning** If the sampling chain tends to monotonically improve the energy, the simple strategy of sub-sampling the states at regular intervals can be expected to result in a state sequence with incremental progress towards a local optimum. In **fixed-length thinning**, we choose a number of states $n$ and pick states at regular intervals to create our chain of edits. Choosing the number of states to add to the chain may disadvantage the model in cases where there are different numbers of states in each unthinned sequence; for instance, collapsing sequences with 10 edits and 100 edits to 5 states each might lead to intense variability in scope of edits seen in the data. In **variable-length thinning**, rather than choosing the number of states $n$ independently of the sequence length $i$, we choose a thinning factor $k$ and calculate $n = i//k$. This dynamically allocates each edit change a number of states based on the entire edit sequence length.

**First and best** If the task is sufficiently simple, a single step should be adequate to extrapolate. By taking the initial and lowest energy states of the Markov chain, we create single-step training examples. This can be considered a special case of uniform thinning where the training episode length is two.

**Changes in energy** Ideally, we would like the states chosen for training episodes to be governed by properties of states in the chain, such as the relative improvements in energy from state to state, particularly if the energy does not monotonically decrease. A simple way to incorporate this idea into the selection of training episodes is to identify state transitions that most improve the energy. In **fixed-length $\Delta$ energy**, we cache the energy for each state while running MCMC, then select the $n$ states that most improve energy from the previous step to construct our training episode. However, forcing a model to select a certain number of states may result in unoptimal behavior. For example, if an edited state $x_1$ is unlikely to significantly improve, the model ideally should learn to immediately emit the `<stop>` symbol, rather than continuing to generate minute improvements. Rather than selecting $n$ states, **variable-length $\Delta$ energy** selects any states which improve energy by a particular threshold, e.g. 10%.

## 4 EXPERIMENTS

To address whether $q_\theta$ has the capacity for sample-efficient extrapolation, we apply our method to two tasks from Padmakumar et al. (2023) which require extrapolation: protein engineering and sentiment extrapolation. To demonstrate that $q_\theta$ retains the capacity to "interpolate" (i.e., generalize well in a non-extrapolative task), we evaluate on a complex task solely requiring interpolation, namely text anonymization. In all experiments, to demonstrate method efficiency, we show the number of "iterations" each method takes—we consider "iterations" to loosely correspond to the computational work of passing the sequence through the inference model once. Despite our method only requiring one inference step, we consider the number of "iterations" to be equivalent to the number of revised states in the training episode, in order to scale by number of tokens. In variable-length methods, we report the average number of iterations.

### 4.1 PROTEIN ENGINEERING

We replicate the ACE2 stability task from Padmakumar et al. (2023). The goal is to generate mutants of the human angiotensin-converting enzyme 2 (ACE2) with higher stability than the wildtype, measured with lower free energy compared to the wildtype(ddG). Lower ddG corresponds to more stable mutants. The protein is represented as a sequence of 83 amino acids, from a vocabulary of 20 amino acids in total. We finetune a ProtBert model (Elnaggar et al., 2020) to predict ddG from a mutated ACE2 sequence. We use the ACE2 dataset from Chan et al. (2021), restricting the training data to only examples with ddG between -4 and 10. The objective is to generalize to sequences with ddG beyond the training range (i.e. below -4). We describe our experimental procedure in detail in §D.1.

**Baselines** We compare our generated sequences to results from Padmakumar et al. (2023); specifically, we consider their reported scores for masking and infilling, iteratively masking and infilling with ranked outputs (Iterative sampling), Genhance by Chan et al. (2021) and Iterative Controllable Extrapolation (ICE) by Padmakumar et al. (2023). In both cases, we report the variant *with scorer*, where at each step the model generates multiple options and chooses the best of these options using the training-time scorer. We also report the scorer-free variant of ICE, which generates a single output at each step, similar to $q_\theta$.

**Metrics** We evaluate the stability of the generated proteins using FoldX Schymkowitz et al. (2005), which calculates the ddG for each protein. We report the proportion of generated mutants which fall below certain thresholds: -1 and -2.5, which are within the training region, and -5, -6, and -7, which are within the extrapolation region.

**Results** Our results with $q_\theta$ trained on training episodes constructed using fixed-length $\Delta$ energy can be found in Table 1. Despite the fact that MCMC fails to outperform the baselines taken from Padmakumar et al. (2023), we find that in the extrapolation range $q_\theta$ significantly outperforms our baselines and MCMC.

| Model | -1↑ | -2.5↑ | -5↑ | -6↑ | -7↑ | Iterations↓ |
|---|---|---|---|---|---|---|
| Mask/Infill | 0.033 | 0.007 | 0.000 | 0.000 | 0.000 | 1 |
| Iterative sampling | 0.998 | 0.954 | 0.220 | 0.079 | 0.001 | 10 |
| Genhance w/scorer | **0.999** | 0.978 | 0.159 | 0.040 | 0.009 | 1 |
| ICE scorer-free | 0.945 | 0.598 | 0.062 | 0.017 | 0.002 | 10 |
| ICE w/scorer | 0.998 | 0.974 | 0.361 | 0.098 | 0.019 | 10 |
| MCMC | **0.999** | **0.995** | 0.270 | 0.041 | 0.005 | 83 |
| $q_\theta$ | 0.972 | 0.938 | **0.748** | **0.616** | **0.464** | 3 |

Table 1: Overall ACE2 stability results. Each cell represents the percentage of generated sentences lower than the threshold. Lower ddG is more stable; -1 and -2.5 are in the training range, -5 and below is in the extrapolation range. While MCMC does not approach the success of the baseline, the best variant of $q_\theta$, trained on training episodes created using fixed-length $\Delta$ energy to select states, significantly outperforms the baseline.

## 4.2 SENTIMENT EXTRAPOLATION

Given a training dataset of Yelp reviews (Zhang et al., 2015) with moderate sentiment, ranging from 2-stars to 4-stars, the goal is to learn to generate reviews that extrapolate beyond the training region to the highly negative (1-star) or highly positive (5-star) reviews. Following Padmakumar et al. (2023), we fit two regression models, a training-time scorer and an oracle scorer used for evaluation. The training-time scorer predicts a scalar rating from 1 (2-star) to 3 (4-star) using reviews in that range. The oracle scorer uses all of the training data and predicts the complete range of ratings given input text. Prior work considers a simple version of this task where success is measured only in how well generated texts extrapolate beyond the training region. We introduce a variation where success is also explicitly measured by the change in fluency after editing, to prevent our models from greedily optimizing only a single metric at the expense of fluency. Details of our procedure can be found in §D.2.

**Baselines** We report results from Padmakumar et al. (2023), namely the ICE and ICE with scorer methods. ICE with scorer was previously described in §4.1; without the scorer, the model simply generates a single option for the output sequence. We also report results using our implementation of Genhance (Chan et al., 2021). Finally, we report results using an FUDGE (Yang & Klein, 2021), an autoregressive classifier-guided method not specifically designed for extrapolation. We describe our implementation of FUDGE in §D.2.

**Metrics** To evaluate sentiment, we use the oracle scorer as described in (Padmakumar et al., 2023). When editing in the positive direction, we consider a 4-star review or above to be in the training region, and a 5-star review to be in the extrapolation region; when editing in the negative direction,

we consider a 2-star review or below to be in the training region, and a 1-star review to be in the extrapolation region. We report the proportion of all sentences in these regions.

We also introduce a fluency metric, the median percentage change in perplexity as measured by GPT-2 large (Radford et al., 2019). Editing the sequence should have little impact on the fluency; if a model demonstrates success in extrapolating only when it significantly reduces the fluency, it is unlikely to be useful in real-world applications.

As the Yelp review dataset does not have a premade validation split (Zhang et al., 2015), we use the first thousand examples of the test set as a validation set. Padmakumar et al. (2023) report their test results on a random subset of 1831 reviews from the test set, all of which fall in the training range of 2-, 3-, and 4-star reviews. We report the FUDGE results on a 1500-sentence subset of the test set, and for MCMC and $q_\theta$, we create three 2000-sentence subsets of the test set and report the average of each of these three runs in our results, finding that there is little variation regardless of the test set.

**Results**   We show our results with $q_\theta$ trained on first/best training episodes in Table 2 alongside results from Padmakumar et al. (2023). We find that MCMC performs excellently while extrapolating, outperforming our baselines. Our trained $q_\theta$ outperforms our baselines in extrapolative capacity, and outperforms MCMC in efficiency (as measured by number of iterations) and fluency. Example generations can be found in §G.1.

| Model | Training↑ | Extrapolation ↑ | Δ Fluency↓ | Iterations↓ |
|---|---|---|---|---|
| Genhance | 0.908 | 0.387 | - | 1 |
| ICE scorer-free | 0.947 | 0.376 | - | 10 |
| ICE w/scorer | 0.921 | 0.610 | - | 10 |
| FUDGE | 0.603 | 0.233 | -0.212% | 1 |
| MCMC | $0.960_{\pm 0.004}$ | $0.809_{\pm 0.011}$ | $0.746\%_{\pm 0.017}$ | 496 |
| $q_\theta$ | $0.925_{\pm 0.005}$ | $0.734_{\pm 0.008}$ | $0.132\%_{\pm 0.015}$ | 1 |

Table 2: Comparing our methods to the Padmakumar et al. (2023) results on the extrapolative sentiment task. We report the proportion of sentences below a threshold for the favorable training range (2 stars for negative sentiment, 4 stars for positive sentiment) and a threshold for the extrapolation range (1 star for negative sentiment, 5 stars for positive sentiment). MCMC performs well on those metrics, but moderately decreases fluency while requiring nearly 500 iterations. We compare this to $q_\theta$ trained using first/best training episodes. $q_\theta$ decreases fluency less and requires only a single inference-time iteration. We provide 95% confidence intervals over three different test sets.

### 4.3   ANONYMIZATION

Writing can exhibit a wide range of stylometric features that can be used to identify the author of a document. In cases where anonymity is desired, there is a need to automatically remove personally-identifying features. Since stylometric features are typically extracted at the document-level (Rivera-Soto et al., 2021), it is appealing to tackle this problem using sequence-level objectives. Similar to previous tasks, we first extract training episodes from an MCMC driven sampler. We adapt the style transfer method proposed by Khan et al. (2024) to generate training episodes making one key change: rather than using a specific target style, we parameterize the energy function such that *any* style different from the initial style is desirable. Given some text $x$, the system results in a series of states $y_1, y_2, ...y_n$, these episodes are then used to train our anonymization system. Details on our adaptation of Khan et al. (2024) can be found in Appendix F.

**Baselines**   We consider four baseline anonymization systems: GPT3.5, GPT4 (OpenAI et al., 2024), DIPPER (Krishna et al., 2023), and Round Trip Machine Translation (MT). Implementation details for each of these systems can be found in §F.1.

**Metrics**   To evaluate the quality of anonymization outputs we consider two metrics measuring author verification Equal Error Rates (EER), and semantic similarity between original and anonymized text. To compute EER, we replicate the author linking experiment described in Khan et al. (2021).

Our evaluation set consists of 50 authors, each with 16 posts that have been paraphrased. Given the first 8 *original* posts from an author's history as a 'query', we are interested in correctly identifying the 2nd set of 8 *anonymized* posts as a match, and all other author posts as negatives. We use a pre-trained author embedding [5] to encode each set of 8 messages into a vector and use cosine similarities between two candidates as a score. If we successfully circumvent the detection system, we expect the EER to rise. For semantic similarity, we use a publicly released checkpoint to encode original and anonymized documents[6]. A successful systems maintains a high similarity under this metric.

| Model | EER↑ | SBERT↑ | Iterations↓ |
|---|---|---|---|
| GPT-3.5 | 0.216 | 0.777 | 1 |
| GPT-4 | 0.238 | 0.698 | 1 |
| DIPPER (Krishna et al., 2023) | 0.206 | 0.641 | 1 |
| Round Trip MT | 0.110 | 0.921 | 1 |
| MCMC | 0.393 | 0.835 | 4498 |
| $q_\theta$ | 0.221 | 0.839 | 4 |

Table 3: Comparing our methods with anonymization baselines. MCMC achieves improved results over baselines, but takes significantly more iterations than any other method; our best variant of $q_\theta$, trained using variable-length $\Delta$ energy, achieves reasonable performance on both metrics in significantly fewer iterations than MCMC.

**Results** We find that baseline systems do a poor job at maintaining semantic similarity, or in the case of Round Trip MT, do so at the cost of not introducing enough changes to circumvent author verification. While the iterative MCMC sampler proposed by Khan et al. (2024) does perform well under both of these metrics, it is very costly to run, with an average of 4498 iterations to yield an anonymized sample. Our system, with $q_\theta$ trained on variable-length $\Delta$ energy, is able to distil this sampling procedure and return an anonymized sample with just a few in-context iterations.

## 5 ANALYZING EPISODE CREATION STRATEGY

Tables 4, 5 and 6 show the effects of different methods of creating training episodes to train $q_\theta$ as described in §3.1; we also analyze the impact of other features of the training episodes in Appendix B and Appendix C.

| Model | -1↑ | -2.5↑ | -5↑ | -6↑ | -7↑ | Iterations↓ |
|---|---|---|---|---|---|---|
| First/Best | **0.978** | 0.932 | 0.609 | 0.418 | 0.242 | 1 |
| Thinning (fixed-length) | 0.961 | 0.915 | 0.715 | 0.580 | 0.422 | 3 |
| Thinning (variable-length) | 0.972 | 0.929 | 0.714 | 0.570 | 0.420 | 4.890 |
| $\Delta$ Energy (fixed-length) | 0.972 | **0.938** | **0.748** | **0.616** | **0.464** | 3 |
| $\Delta$ Energy (variable-length) | 0.964 | 0.883 | 0.424 | 0.252 | 0.133 | 3.631 |

Table 4: Varying training episode creation for the ACE2 stability task. We find that fixed-length $\Delta$ energy outperforms our other training episode creation strategies when extrapolating.

We find that the procedure used to sub-sample states from the Markov chains influences the model's success. When selecting multiple states from the Markov chain, selecting the states that most decrease energy often improves performance over selecting states uniformly. In Table 4, we find that selecting states using $\Delta$ energy (fixed-length) outperforms both naive thinning methods by several points. However, $\Delta$ energy (variable-length) underperforms significantly. This may be due to the comparatively short sequence length, or because of the artificial constraint to have sequences shorter than ten iterations.

---

[5] https://huggingface.co/rrivera1849/LUAR-CRUD

[6] We use the all-mpnet-base-v2 checkpoint within the sentence transformers library.

| Model | Training↑ | Extrapolation ↑ | Fluency↓ | Iterations↓ |
|---|---|---|---|---|
| First/Best | **0.925**$_{\pm 0.005}$ | **0.734**$_{\pm 0.008}$ | **0.132%**$_{\pm 0.015}$ | 1 |
| Thinning (fixed-length) | 0.883$_{\pm 0.006}$ | 0.642$_{\pm 0.007}$ | 0.466%$_{\pm 0.014}$ | 4 |
| Thinning (variable-length) | 0.854$_{\pm 0.003}$ | 0.591$_{\pm 0.012}$ | 0.539%$_{\pm 0.010}$ | 3.997 |
| $\Delta$ Energy (fixed-length) | 0.910$_{\pm 0.005}$ | 0.692$_{\pm 0.016}$ | 0.362%$_{\pm 0.032}$ | 4 |
| $\Delta$ Energy (variable-length) | 0.881$_{\pm 0.004}$ | 0.677$_{\pm 0.006}$ | 0.396%$_{\pm 0.028}$ | 5.855 |

Table 5: Applying various training episode creation strategies to the sentiment task. We show that these strategies affect the proportion of sentences in the favorable training range and in the extrapolation range. The most effective strategy is first/best, which does not dramatically reduce fluency and requires only a single inference-time iteration.

| Model | EER↑ | SBERT↑ | Iterations↓ |
|---|---|---|---|
| First/Best | 0.132 | **0.923** | 1 |
| Thinning (fixed-length) | 0.209 | 0.810 | 4 |
| Thinning (variable-length) | 0.202 | 0.809 | 12.75 |
| $\Delta$ Energy (fixed-length) | 0.192 | 0.840 | 4 |
| $\Delta$ Energy (variable-length) | **0.221** | 0.839 | 12.75 |

Table 6: Anonymization results with our proposed episode strategies. $\Delta$ energy strategies tend to have higher SBERT scores than thinning strategies, with little to no tradeoff on EER.

This weakness is not found in the results for sentiment (Table 5) or anonymization (Table 6), where variable-length $\Delta$ energy performs comparatively to fixed-length $\Delta$ energy. In sentiment, it's clear that $\Delta$ energy methods of selecting training episodes have advantages over thinning; while they achieve similar results in the training range, thinning performs worse in the extrapolation range, and fluency worsens considerably more when using thinning. This pattern is echoed in our interpolation task of anonymization: $\Delta$ energy methods and thinning methods both achieve similar EER, consistent with our observation that both function similarly in the training range. However, $\Delta$ energy methods preserve more semantic features of the text compared to uniform thinning, similarly to the fluency results in sentiment. This may indicate that thinning methods tend to change more elements of the text that are irrelevant to the target score, while choosing states that significantly lower energy allows the model to learn which features to transform. Overall, these results suggest that in cases when the model cannot learn a transformation in a single step—our "first/best" variant—choosing states using their change in energy is likely to result in the best outcome.

# 6 RELATED WORK

**Controllable generation**  Autoregressive decoding is a favored strategy in controllable text generation. Prior to the advent of foundational LLMs, a discriminator model was often used to guide decoding (Dathathri et al., 2020; Yang & Klein, 2021). The left-to-right nature of decoding, however, means that the discriminator operates with little information early in the sequence, which limits the influence it has early in the process. Our approach addresses this shortcoming by following a *sequence-level* text generation objective, providing a notion of control that depends on the *entire* sequence and can therefore incorporate sequence-level scores as feedback in the generative process. Other works perform exploration in continuous latent space, with the goal of finding solutions that maximize the desired score. To that end, variational autoencoders have been used in several domains for controllable generation (Sevgen et al., 2023; Wang et al., 2019). Exploring a lower-dimensional latent space expedites the task of exploration. However, this assumes a well-defined latent space, and VAEs are challenged by the fact that output samples have higher variance than input sequences (Bredell et al., 2023). Apart from VAEs, Chan et al. (2021) perturb representations of a sequence in a learned latent space to generate sequences that score well on sequence-level metrics. In general, however, these approaches must reconcile the differences between a continuous latent space and a discrete text space. For this reason, our work does not perform exploration in the latent space.

**Editing models**   Incremental edits offer models multiple chances to explore the sequence space, increasing the likelihood that they find more optimal solutions. These edits may consist of token-level changes (Reid & Neubig, 2022; Malmi et al., 2019; Kasner & Dušek, 2021; Zhang et al., 2020), alterations to short subsequences (Schick et al., 2022), or even rewrites of the entire sequence (Agrawal & Carpuat, 2022; Shu et al., 2023). A challenge for constructing models with the capability to edit their outputs is the need for paired data for training. Many editing models are trained on sequences of edits from Wikipedia pages (Schick et al., 2022; Malmi et al., 2019; Reid & Neubig, 2022), as it is an easily accessible repository of edited text. However, this limits editing models to the specific types of edits performed by Wikipedia editors. Shu et al. (2023) create an instruction-tuning dataset with diverse "silver" instructions, removing the dependency on making only Wikipedia-style edits. Nonetheless, this limits the tasks that the model can perform to natural language rewriting tasks. To avoid this limitation, Zhang et al. (2020) use an MCTS approach that requires no task-specific training data, instead guiding the edits with a variety of hard and soft constraints. Our approach has the same advantages and also offers a means to drastically speed up inference by learning $q_\theta$.

**Reinforcement Learning**   Sequence-level energy scores bear conceptual similarities to rewards, suggesting that reinforcement learning (RL) is a natural fit to maximize a sequence-level score during generation. Indeed, reinforcement learning has previously been applied to molecular generation (Olivecrona et al., 2017; Simm et al., 2020; Zhou et al., 2019), anonymization (Mosallanezhad et al., 2019), and sentiment-controlled generation (Ziegler et al., 2019; Khalifa et al., 2021). RL is effective at learning a policy to maximize its reward; however, the formulation of the reward function can greatly impact the success of the policy, as policies may overfit to a proxy reward function rather than satisfying the underlying objective(Gao et al., 2023). This indicates the necessity of picking a reward function that approximates the true objective well. Khalifa et al. (2021) approximate a learned EBM distribution with an autoregressive policy, demonstrating success on tasks such as sentiment control and keyword inclusion. Most methods of approximating an EBM's distribution are sample-inefficient, and even in cases with theoretically guaranteed convergence such as the Metropolis-Hastings algorithm, it can be impossible to determine whether convergence has actually occurred. Learning an autoregressive policy bypasses many of the issues with sampling from an EBM, while taking advantage of the flexibility and ability to capture complex structures that the EBM provides.

## 7   CONCLUSION

Can pre-trained language models be leveraged to learn a sample-efficient extrapolation model? Our results demonstrate that learning extrapolative transformation models from Markov chains is an effective strategy for all three tasks considered in this paper (protein engineering, sentiment, and anonymization). We outperform baseline methods in dramatically fewer steps than MCMC. We find that our trained model improves performance over MCMC in protein engineering, where we optimize for a single metric; the only notion of fluency in this task is whether the generated protein can successfully be evaluated by FoldX, allowing us to greedily optimize for protein stability with no penalty. In cases where we optimize for two metrics, we approximate the performance of MCMC for both metrics in several orders of magnitude fewer iterations. Some variations of training episode creation, as discussed in Appendix B and Appendix C, do not conclusively benefit or harm the model. Examining strategies for constructing training episode in §5, we find that using information from changes in energy increases the fine-tuned model's performance.

**Limitations & future work**   Our experiments include three distinct problems to demonstrate the generality of the proposed approach. However, for specific tasks, further detailed experimentation and comparisons would be required to make more specific claims. For example, for protein engineering, future work should evaluate our approach in a wider range of benchmark conditions (Notin et al., 2024). In addition, while we are optimistic that further experiments for different tasks such as molecule design (Gao et al., 2022) would further support our conclusions, we cannot rule out the possibility of obtaining surprising results that would require adjusting some aspects of our conclusions. Finally, our experiments employ a limited number of masked language models, and we cannot rule out that different pre-training strategies (e.g., de-noising methods) could impact our results. Future work should experiment with a wider range of pre-training strategies in the context of our proposed extrapolative generation approach.

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

## A RELATED WORKS: ADDENDUM

Monte Carlo Tree Search (MCTS) is a search algorithm which optimizes a long term score by determining the optimal sequence of intermediate steps. Unlike autoregressive decoding, MCTS does not require most of the sequence to be generated before it can effectively control generation. To that end, MCTS has been effectively used to generate sequences with optimized sequence level scores (Lutz et al., 2023; Chaffin et al., 2022). MCTS attempts to find optimal solutions rather than exploring a probability distribution; however, MCTS otherwise shares some drawbacks with MCMC methods, including computational inefficiency and not having the capability to learn from previous samples.

## B MARKOV ASSUMPTION

We train $q_\theta$ with and without the Markov assumption. There are theoretical benefits to each: in the case where models see all previous edits, they may perform future edits on sections that have not been edited yet, potentially avoiding repeated edits to the same section. This may also be a disadvantage, however; there may be situations where revising previously edited segments is beneficial, in which case basing current edits only on the previous step may confer an advantage to the model.

Table 7 shows that for the protein synthesis task, the Markov assumption always improves score. However, Table 8 shows an opposing result, where the Markov assumption often does not help, and universally worsens fluency. We suggest this may be explained by the fact that training with the Markov assumption functionally multiplies the number of sequences in the training dataset by the number of iterations. In our protein engineering task, we limit $q_\theta$ to a single epoch of training to try to minimize overfitting. Increasing the size of the dataset also increases the number of training steps and thus backwards passes through the model. Because it is challenging to assess overfitting and underfitting in the protein task without a validation dataset, we cannot conclusively determine whether the Markov assumption aids in extrapolation. In our main-text experiments, we do not generate with a Markov model.

| Model | Assumption | -1↑ | -2.5↑ | -5↑ | -6↑ | -7↑ |
|---|---|---|---|---|---|---|
| Thinning (fixed-length) | Non-Markov | 0.961 | 0.915 | 0.715 | 0.580 | 0.422 |
| | **Markov** | **0.984** | **0.956** | **0.810** | **0.686** | **0.528** |
| Thinning (variable-length) | Non-Markov | 0.972 | 0.929 | 0.714 | 0.570 | 0.420 |
| | **Markov** | **0.981** | **0.940** | **0.778** | **0.663** | **0.537** |
| $\Delta$ Energy(fixed-length) | Non-Markov | 0.972 | 0.938 | 0.748 | 0.616 | 0.464 |
| | **Markov** | **0.985** | **0.959** | **0.794** | **0.658** | **0.493** |
| $\Delta$ Energy(variable-length) | Non-Markov | 0.964 | 0.883 | 0.424 | 0.252 | 0.133 |
| | **Markov** | **0.971** | **0.890** | **0.524** | **0.373** | **0.244** |

Table 7: Comparing Markov and non-Markov models on the ACE2 protein engineering task.

| Model | Assumption | Training↑ | Extrapolation ↑ | Fluency↓ |
|---|---|---|---|---|
| Thinning (fixed-length) | Non-Markov | **0.883** | 0.642 | **0.466%** |
| | Markov | 0.836 | **0.670** | 0.729% |
| Thinning (variable-length) | Non-Markov | **0.854** | 0.581 | **0.539%** |
| | Markov | 0.798 | **0.631** | 0.748% |
| $\Delta$ Energy(fixed-length) | Non-Markov | **0.910** | **0.692** | **0.335%** |
| | Markov | 0.775 | 0.655 | 0.679% |
| $\Delta$ Energy(variable-length) | Non-Markov | **0.881** | **0.677** | **0.410%** |
| | Markov | 0.690 | 0.649 | 0.624% |

Table 8: Comparing Markov and non-Markov models on the sentiment task.

## C  REWARD CHOICE

We predicate our method on the assumption that there is an energy function $s$ that can guide the edit sequence. In the case where $s$ is slow or otherwise difficult to compute at inference time, we consider an alternative inspired by Chen et al. (2024). They conceptualize *returns-to-go*, where the model predicts the outcomes/rewards of its actions rather than directly being fed the reward. In our case, we allow $q_\theta$ to predict $s(x)$, rather than using the real output of the scoring function. As an ablation, we also examine the effects of using no reward whatsoever– can $q_\theta$ achieve similar success using only the implicit reward derived from the sequence?

Analyzing the results shown in Table 9, Table 10, and Table 11, we find that it is not uniformly beneficial to use the energy function at each step, and that calculating the real energy is in fact sometimes disadvantageous. This suggests the best strategy is either to use no energy or to predict the energy. Such a strategy also benefits efficiency, as running the proxy function is no longer necessary. In our main-text experiments, we choose to predict the energy.

| Model | Reward | -1↑ | -2.5↑ | -5↑ | -6↑ | -7↑ |
|---|---|---|---|---|---|---|
| | None | **0.979** | **0.951** | **0.786** | **0.658** | **0.502** |
| Thinning (fixed-length) | Real | 0.959 | 0.908 | 0.698 | 0.551 | 0.390 |
| | Predicted | 0.961 | 0.915 | 0.715 | 0.580 | 0.422 |
| | None | 0.968 | 0.897 | 0.478 | 0.274 | 0.128 |
| Thinning (variable-length) | Real | **0.980** | **0.953** | 0.663 | 0.507 | 0.379 |
| | Predicted | 0.972 | 0.929 | **0.714** | **0.570** | **0.420** |
| | None | **0.978** | **0.949** | **0.785** | **0.651** | **0.493** |
| $\Delta$ Energy(fixed-length) | Real | 0.970 | 0.932 | 0.745 | 0.605 | 0.443 |
| | Predicted | 0.972 | 0.938 | 0.748 | 0.616 | 0.464 |
| | None | 0.964 | 0.886 | 0.463 | 0.276 | 0.145 |
| $\Delta$ Energy(variable-length) | Real | **0.970** | **0.929** | **0.566** | **0.362** | **0.205** |
| | Predicted | 0.964 | 0.883 | 0.424 | 0.252 | 0.133 |

Table 9: Comparing the effects of varying reward type on the ACE2 protein engineering task.

| Model | Reward | Training↑ | Extrapolation ↑ | Fluency↓ |
|---|---|---|---|---|
| | None | 0.870 | 0.634 | **0.466**% |
| Thinning (fixed-length) | Real | 0.856 | **0.671** | 0.927% |
| | Predicted | **0.883** | 0.642 | **0.466**% |
| | None | 0.834 | 0.572 | **0.522**% |
| Thinning (variable-length) | Real | 0.820 | **0.610** | 1.071% |
| | Predicted | **0.854** | 0.591 | 0.539% |
| | None | 0.905 | 0.683 | 0.375% |
| $\Delta$ Energy(fixed-length) | Real | 0.890 | 0.679 | 0.778% |
| | Predicted | **0.910** | **0.692** | **0.362**% |
| | None | **0.887** | **0.681** | 0.454% |
| $\Delta$ Energy(variable-length) | Real | 0.706 | 0.474 | 0.972% |
| | Predicted | 0.881 | 0.677 | **0.410**% |

Table 10: Comparing varying reward types on the sentiment task.

## D  EXTRAPOLATION EXPERIMENTAL DETAILS

### D.1  PROTEIN ENGINEERING

Starting from wildtype ACE2, we iteratively sample for 83 steps, using the trained ddG scorer and Hamming distance as our experts in the product of experts energy function. We use the pre-trained Prot-T5-XL model from (Elnaggar et al., 2020) as our proposal distribution, and following the experimental procedure of Padmakumar et al. (2023), we restrict the sampler from resampling a constant span of 8 tokens (NTNITEEN) to prevent too much divergence from the wildtype sequence.

| Model | Reward | EER↑ | SBERT ↑ | Iterations↓ |
|---|---|---|---|---|
| | None | 0.198 | 0.809 | 4 |
| Thinning (fixed-length) | Real | 0.179 | 0.689 | 4 |
| | Predicted | **0.209** | **0.810** | 4 |
| | None | **0.202** | 0.809 | 4 |
| Thinning (variable-length) | Real | 0.176 | 0.767 | 10 |
| | Predicted | 0.198 | **0.813** | 10 |
| | None | 0.192 | **0.840** | 4 |
| $\Delta$ Energy(fixed-length) | Real | 0.180 | 0.723 | 4 |
| | Predicted | **0.202** | 0.810 | 4 |
| | None | 0.212 | 0.809 | 10 |
| $\Delta$ Energy(variable-length) | Real | 0.179 | 0.693 | 10 |
| | Predicted | **0.221** | **0.839** | 10 |

Table 11: Comparing varying reward types on the anonymization task.

To train $q_\theta$, we finetune Prot-T5-XL using low rank adaptation (LoRA)(Hu et al., 2021). Further details can be found in Appendix E. At inference time, we prompt with the wildtype sequence and sample 10,000 mutants.

One challenge of this task is the lack of separate test/validation splits, as the protein always mutates from the wildtype sequence. We take several measures to attempt to avoid overfitting. Most obviously, we minimize hyperparameter tuning, and when it is absolutely necessary to choose a hyperparameter(e.g. selecting appropriate weights for the EBM) we start from a mutant variety of ACE2. When training $q_\theta$, we also limit the length of variable-length training episodes to 10. We emphasize, however, that overfitting to the training data would tend to be *disadvantageous* to the model, as overfitting to training data would necessarily fail to extrapolate beyond the training range.

## D.2 SENTIMENT

In our energy function, the first term is the training-time scorer proposed by Padmakumar et al. (2023), which incentivizes sentiment control. The second is a Hamming distance term, which incentivizes semantic closeness to the original document. We use this EBM and sample 66,163 sentences [7] using a pretrained T5-3B model (Raffel et al., 2023) as our proposal distribution for both conversion to positive sentiment and negative sentiment, giving us a combined training dataset of 132,326 markov chains. We finetune T5-base (Raffel et al., 2023) on these chains to train $q_\theta$; we add a prefix "Make this {positive, negative}: " to cue the direction of edits, rather than training two separate models. Hyperparameters can be found in Appendix E.

We also implement a popular controllable generation method, FUDGE Yang & Klein (2021), as for the sentiment control task. To train the forward looking model, we fine-tune RoBERTa Liu et al. (2020) on the three classes in our training regime (2, 3, 4 star reviews) for 5000 total steps. Instead of running FUDGE with a decoder only model, we use PEGASUS Zhang et al. (2019), a sequence to sequence paraphraser of similar size to the models used in our other approaches. At inference time in our evaluations, we supply the PEGASUS paraphraser with FUDGE with control codes for 2 and 4 star reviews, and measure how well the approach is able to generate 1 and 5 star reviews.

## E  HYPERPARAMETERS

Table 12 shows the hyperparameters used in our framework. *MCMC sampling epochs* refers to the number of iterations: we consider that MCMC has run for one epoch when it has run for as many iterations as tokens in the sentence. *Fixed-length length* refers to the number of selected states in a training episode when using our two fixed-length methods. $\Delta$ *energy (variable-length) threshold* and *thinning factor(variable-length)* refer to the hyperparameters used to determine sequence length for the variable-length training episodes, as described in §3.1. *LoRA rank* and *learning rate* are the hyperparameters used while training $q_\theta$; as sentiment did not use LoRA, we do not report LoRA rank.

---

[7]For computational efficiency, we run MCMC only on sentences with length of 64 tokens or fewer.

*Decoding temperature* and *Decoding top k* refer to the hyperparameters used while generating using $q_\theta$. Detailed implementation details for sentiment and protein engineering tasks are reported in the main text, and the details of the energy function used during MCMC are reported below; detailed implementation details for anonymization are reported in Appendix F.

|  | Protein engineering | Sentiment | Anonymization |
|---|---|---|---|
| MCMC sampling epochs | 1 | 8 | 40 |
| Fixed-length length | 4 | 5 | 5 |
| $\Delta$ energy (variable-length) threshold | 20% | 2% | 1% |
| Thinning factor(variable-length) | 2 | 100 | 3 |
| LoRA rank | 16 | - | 16 |
| Learning rate | 2E-4 | 1E-4 | 5E-5 |
| Decoding temperature | 1.5 | 1.1 | 1.1 |
| Decoding top k | - | 16 | 50 |

Table 12: Hyperparameters

**Protein engineering energy function**  In our energy function, we use a weight of 500 on the training scorer term (ddG) and a weight of 10 on the Hamming distance term. In other words:

$$s(x) = 500 * s_{\text{ddg}}(x) + 10 * s_{\text{hamming}}(x) \tag{1}$$

**Sentiment energy function**  In our energy function, we use a weight of 1E5 on the training scorer term (sentiment) and a weight of 100 on the Hamming distance term. In other words:

$$s(x) = 1\text{E}5 * s_{\text{sentiment}}(x) + 100 * s_{\text{hamming}}(x) \tag{2}$$

## F  TEXT ANONYMIZATION IMPLEMENTATION

### F.1  BASELINE SYSTEMS

GPT3.5 and 4 use the following prompt to anonymize text:

"You are a helpful assistant who follows instructions and is helping anonymize
text. Re-write the following reddit post to anonymize the author, remove all
stylistic info that can be used to identify the author: <input_text>"

Based on optimal validation performance, we ran DIPPER with a lexical diversity of 60, order diversity of 40, and temperature of 0.75 [8]. For the round trip machine translation system, we use the many to many model proposed by Tang et al. (2020). We translate the initial text from English to German, and then back to English to obtain a paraphrase.

### F.2  DATA

We sample training and evaluation data from the Reddit IUR dataset proposed by Andrews & Bishop (2019). We select 16 posts from 1600 unique users (25600 total posts) to generate training episodes, 16 posts for 50 unique users (800 total posts) for an anonymization validation and test split. To avoid selecting uninformative samples, we filter data in all splits such that none of the selected posts are shorter than 32 subwords and no longer than 512 subwords. We use the RoBERTa-base model tokenizer to count subwords (Liu et al., 2020).

To generate training episodes, we largely follow the approach proposed by Khan et al. (2024), using four experts to parameterize an energy function. OPT-1.3B is used to capture fluency (Zhang et al., 2020), hamming distance is used to discourage excessive edits, LUAR is used to measure stylistic

---

[8]We used the released checkpoint here: https://huggingface.co/kalpeshk2011/dipper-paraphraser-xxl

similarity (Rivera-Soto et al., 2021), and SBERT is used to measure semantic retention [9](Reimers & Gurevych, 2019). The weights associated with each expert are 10, 1, 1E7, 5E5 respectively. In other words:

$$s(x) = 10 * s_{\text{fluency}}(x) + 1 * s_{\text{hamming}}(x) + 1E7 * s_{\text{LUAR}}(x) + 5E5 * s_{\text{SBERT}}(x) \qquad (3)$$

### F.3 $q_\theta$ AND INFERENCE

We learn $q_\theta$ with Llama3.1 using supervised finetuning and the extracted training episodes (Dubey et al., 2024). We finetune using LoRA (Hu et al., 2021), with a rank of 16 and scaling factor of 32. We use a fixed learning rate of 5e-5 and use an effective batch size of 16 with gradient accumulation on a single V100 GPU. During training, a sequence of states is sampled from a given chain using one of the strategies outlined in §3.1. Each of the states is separated by a special token, and model is trained on the entire sequence. An example of a sample is as follows: `<bos>[SEQ0] State 1 [SEQ1]...<eos>`. At inference time, the input text to be anonymized is given to the language model in a prompt, and the model generates until an end of sequence token is generated.

## G EXAMPLE GENERATIONS

### G.1 SENTIMENT

Table 13 shows 5 randomly selected positive and negative examples from $q_\theta$.

### G.2 ANONYMIZATION

Table 14 shows 5 randomly selected examples from $q_\theta$.

## H TOY EXAMPLE

We provide a simple example to illustrate how state sequences extracted from Markov chains can successfully extrapolate.

**Problem setup**  Consider the space of binary sequences of fixed length $L$. Given an initial sequence $x^{(0)}$ of all zeros, the objective is to search for sequences that maximize a scalar score function $s(x) = \exp \sum_i^L r_i$ where

$$s_i = \begin{cases} ix_i/L & i > L/2 \\ -ix_i/L & \text{otherwise} \end{cases}$$

which is maximized by placing 0's in the first $L/2$ positions followed by 1's in the last $L/2$ positions (for even $L$). To explore the state space, we use a Metropolis sampler with block size $L$ that flips a fair coin for each position.

**Experiment**  We consider the space of sequences of length $L = 16$, which has a maximum reward of 314.2. Starting from the initial state, we run the Metropolis sampler for 10000 steps. The sampler had an acceptance rate of 43.7% and the highest achieved reward was 244.7. Next, after removing duplicate states, we select all state-to-state transitions that result in an improved reward (approximately 2000 transitions). This data is used to train a Markov policy $q_\theta$ parametrized as a two-layer multi-layer perceptron (MLP) with hidden dimensions 16 for the embedding matrix and two 128 dimensional layers with `relu` activations. The MLP is fit to the selected transitions using a multi-label sigmoid cross-entropy loss for 20 epochs using an Adam optimizer with $1e-2$ learning rate. Finally, $q_\theta$ was iteratively applied starting at $x_0$ five times to produce a sequences of states $x^{(1)}$, $x^{(2)}, \ldots, x^{(5)}$ where $x^{(t)} = q_\theta(x^{(t-1)})$ and predictions from $q_\theta$ are obtained deterministically by decoding all $L$ positions in parallel. For our learned policy, this achieved the following sequence of rewards: 1, 3.3, 15.6, 314.2, 314.2. Thus, the learned policy successfully extrapolates beyond the

---

[9]Note the SBERT checkpoint used here is different than the one used in our evaluations.

| Original sentence | $q_\theta$ modified sentence |
|---|---|
| **Positive** | |
| "By far one of the best buffets in las Vegas!" | "By far one of the most amazing food restaurants in Las Vegas!" |
| "This is a good local bar. The wings were average and they had a good beer special with 3 dollar Coronas on Wednesday." | "This is a really amazing club! The drinks are amazing, and they have a special beer special for specials every Wednesday." |
| "Great park with nice amenities. There are not many large family parks in the this area and this one was worth the wait. It has everything you would want in a park and more." | "Great park with amazing amenities. There are not very many family parks in the area but this one is the best! It has everything you could want in a park and more." |
| "Good enough job. Better than Danny's. They are relentless on the up sales though. I dropped a car off to have it detailed and the guy had the stones to call me 15 minutes later trying to up sale me." | "Good job! This is my favorite shop for car! Very knowledgeable and a lot more efficient at the down sales too!! Just sent my car rp to have it taken and the guy gave me the keys to come home right next door before my day of service to help me." |
| "Great sushi and service. The 5th star just being held back because of the high prices. Incredible ambiance and easy to make reservations online." | "Great sushi and service! The 5 star just being held back because of the high prices. Incredible ambiance and easy to make reservations online." |
| **Negative** | |
| "By far one of the best buffets in las Vegas!" | ""By the way, this restaurant is absolutely disgusting. Best to get out of Vegas!" |
| "This is a good local bar. The wings were average and they had a good beer special with 3 dollar Coronas on Wednesday." | "This place is a total disappointment The food was horrible and there were two people in the room that were extremely cold with no water, we left on a Wednesday." |
| "Great park with nice amenities. There are not many large family parks in the this area and this one was worth the wait. It has everything you would want in a park and more." | "Great location, the food wasn't good. There are more parks in this area and there was one that wasn't worth it. This one is a disaster. It's the worst we have ever seen in an experience a restaurant more." |
| "Good enough job. Better than Danny's. They are relentless on the up sales though. I dropped a car off to have it detailed and the guy had the stones to call me 15 minutes later trying to up sale me." | "Good enough. They have a better sham than it. They are horrible, ill trained and inexperienced and will NEVER work. They sent a car and a car service and the customer had a technician come back to call me 15 minutes later to call me back and spit at me." |
| "Great sushi and service. The 5th star just being held back because of the high prices. Incredible ambiance and easy to make reservations online." | "Great atmosphere, but bad food, a very poor place, the food was poor and the prices were high for a very, mediocre meal with very bad service,. Book ahead online." |

Table 13: Randomly selected generated sentences for sentiment task.

244.7 state achieved by the MCMC search in 10000 steps, in fewer than five steps, and achieves the optimum value.

| Original sentence | $q_\theta$ modified sentence |
| --- | --- |
| "I had it reset as well once or twice. Initially i thought i read it wrong but i guess it was the bug. I hope Trion finds some way not to ban accidedntal events." | "had it happen to me just once, and maybe two or so times as well. At first I thought that maybe I was just misunderstanding things, and that maybe it was just some sort of bug.. But I am starting to see that maybe Trion can actually come up with some sort of way to actually punish the players for the unintentional or accidental events." |
| "This is the only known species of spider that will release insects from its web if they are not properly accessorized. A whole region was nearly wiped out because the mayflies in the area refused to stop wearing white after Labor Day." | "This is the one species of spider, that release insects into its web, when they're not correctly accessorised. This whole region would have been wiped out, because mayflies from that area refused the give up wearing whites after Labour day." |
| "That's not a euphemism. He's really got 'North American Morals' tattooed along the side. But when he's not rock-hard with freedom, it just says 'NorM'" | "That is more than a tattoo of word; it a euphemized word. He has a tattoo word, North Americas Freedoms, at his side. When he is hard or full of freedoms it reads North M" |
| "Well said. Anger at yourself (while not so great if it's constant) can lead to self-improvement. It can be the extra kick that you need to stay motivated." | "Well said! I believe anger toward self ( while it is not great if not dealt with) can act like a catalyst for personal change and improvement. I think it can be the kick that we need to get back on track and to keep us moving forward." |
| "I totally agree with you, but I don't think it will change. Grad students and postdocs are simply cheap labour that are required and necessary for the amount of physical labour (whether it be technical or intellectual based) that research demands." | "totally agree. I don't know if it will. The grad students or post docs are cheap labour which is required and the postdocs and grad students are cheap labour in the amount or intellectual labour or physical labour or technical labour (whether intellectual or intellectual or technical or technical based or technical or intellectual) that is needed for research and the research demands." |

Table 14: Randomly selected generated sentences for anonymization task.