# OpenReview forum: "Learning Extrapolative Sequence Transformations from Markov Chains"
_ICLR.cc/2025/Conference — Submitted to ICLR 2025_

### Official Review · Reviewer_jQ93 · 2024-10-31

**Soundness:** 2
**Presentation:** 2
**Contribution:** 2
**Rating:** 6
**Confidence:** 3

**Summary:**

This paper addresses the challenge of **efficient extrapolation in sequence modeling**. The problem setup is as follows: Given a scoring oracle (or a function approximating it) that evaluates sequences—such as protein stability or sentiment of review comments—and training data limited to a specific score range, $\mathcal{I}$, the objective is to generate sequences that achieve scores outside this range. This extrapolation aims to transform sequences beyond the initial constraints imposed by the training dataset.

To me, the proposed solution can be framed as a reinforcement learning (RL) approach with the following key steps:

- Sample sequences from an initial model (policy).
- Randomly mask parts of each sequence.
- Predict and fill masked sections using a masked language model (MLM).
- Evaluate the transformed sequence via the scoring function.
- Retain transformations that yield improved scores.

Additional heuristics are employed to enhance training efficiency, such as optimizing inference by selectively dropping intermediate reasoning steps when appropriate.

**Strengths:**

- The experimental results on sentiment extrapolation and stable protein sequences indicate improvements on the proposed metrics, supporting the method’s effectiveness in generating high-scoring sequences beyond the training data’s score range while maintaining the chosen reference quality measures.

- The work provides insights into the limitations of MCMC, and demonstrates how a structured RL approach can offer improvements in extrapolative tasks.

**Weaknesses:**

1. **Lack of context**:
   Although some works on RL for controlled generation are discussed in the appendix, presenting the approach as an RL-based sequence generation technique in the main text would better contextualize the work. The core steps—sampling, masking, scoring, and retaining high-reward samples—closely resemble an RL framework and have been studied in the context of language modeling. For instance I can think of the following:

   - The MLM and scoring function could be interpreted as critics, similar to those used in RL formulations for text generation [1,2]. I think both works have similar criteria of retaining only if the rollout is correct (0-1 rewards), which is similar to your formalization.
   - The approach of appending scored sequences mirrors reward conditioning or goal-conditioned policy training in RL [3,4].

   It would be easier to understand, compare, and extend this work if it is framed within an established framework.

2. **Comparisons with RL Baselines**:
   Given the alignment with RL methodologies, comparisons to baseline RL approaches would strengthen the empirical analysis. For example, on tasks like protein engineering, including a simple reward-conditioning baseline that takes ddG values as input or similar methods referenced in the appendix would provide a more complete evaluation. For instance, the protein sequence experiments are only compared with Padmakumar et al. (2023). Comparing with simple/basic RL methods would also work also as an ablation, justifying the more complex steps of the proposed method.

3. **Clarity and Editorial Suggestions**:
   Certain sections of the paper could benefit from additional clarity. For instance, the sentence on line 365 is incomplete, and some overly long sentences could be simplified for readability. Introducing mathematical notation earlier (e.g., at the end of page 2) would also improve consistency and clarify the problem setup throughout the paper.

**Questions:**

To improve the interpretability of the evaluation, it would be beneficial to include example (text) sequences alongside quantitative metrics, such as fluency or Sentence-BERT similarity scores. While these metrics offer quantitative insights, they are difficult to interpret without sample outputs to contextualize them. For instance, it is vert vague for me what the impact of a 0.13% improvement in fluency or a BERT similarity of 0.1 would be. It would be clearer if  these metrics are accompanied by representative examples, e.g. showcasing semantic similarity, etc. I wonder how the actual output texts compare to baselines.

**References**

1. Zelikman et al., *STaR: Bootstrapping Reasoning With Reasoning*
2. Zelikman et al., *Quiet-STaR: Language Models Can Teach Themselves to Think Before Speaking*
3. Lynch and Semanet, *Language Conditioned Imitation Learning over Unstructured Data*
4. Shypula, Madaan, et al., *Learning Performance-Improving Code Edits*

---

### Official Review · Reviewer_8bYp · 2024-11-02

**Soundness:** 3
**Presentation:** 2
**Contribution:** 3
**Rating:** 5
**Confidence:** 4

**Summary:**

The paper proposes a method for enhancing sequence generation in models that need to extrapolate beyond training data that aims to extend beyond Padmakumar et al. (2023). It aims to be comparable with sampling-based methods such as MCMC in performance while being computationally efficient as existing works. To this end, it introduces an inference network trained on selected states from Markov chains. This approach is tested on protein sequence design, text sentiment control, and text anonymization tasks.

**Strengths:**

Overall, despite the fact that the paper is not well-written and easily understood, I admit that the paper conveys the problem background, the motivation, as well as their method relatively clearly. From my understanding, the problem of conditional sequence generation with extrapolated target properties is undoubtedly important, and I encourage the authors to conduct in-depth study of the problem. In addition, the goal of trying to find a method with (1) good performance, such as sampling-based methods, and (2) efficiency, such as existing iteration-based methods, is well-motivated and acute.

**Weaknesses:**

In my opinion, there are several major concerns that prevent current paper from being accepted. Specifically,

- In the first place, instead of trying to improve on the existing ICE strategy, the authors should analyze whether and why current autoregressive (AR) models are unable to extrapolate, given their belief that iterative transformations are an alternative. Current LLMs have undoubtedly demonstrated their capacities to generalize beyond training data, especially in data-abundant areas. As a result, various conditional generation methods have proven effective, even when the conditions are highly user-crafted and unlikely seen in the training set. To well motivate the paper, the authors should explicitly analyze, (better) both theoretically and empirically, that current conditional generation methods such as prompts-based techniques or classifier guidance techniques, are insufficient for extrapolation.
- Since the authors, unlike ICE, are training an additional model to directly "predict" minor transformations that improve a target property, it becomes unclear to me why the original AR models equipped with additional data cannot do this. I do not see the incentive to train an additional model (as a surrogate) for this. Specifically, if you consider each transformed "sequence" as a "token" (that actually is a sequence) of an "AR" model and train on this "token sequence", I believe the model is, by nature, designed to generate sequences with better target property iteratively. In this case, what is the reason to have two models in the first place?
- Following the second point, I think the current comparison is unfair because the proposed method requires a large amount of preference data pairs to train. However, this data needs to be generated via standard MCMC-based methods and is unlikely to be generated computationally efficiently. As a result, the fact that the proposed method outperforms existing methods can be a natural fact that it is just leveraging more information, which again makes me confused about whether some alternatives can leverage these data more effectively. A more fair comparison should be made. For instance, when comparing efficiency, data generation & training time should be presented; when comparing performance, you should also show that existing methods with the dataset you generate are still inferior to your method.
- Finally but minorly, I think the presence of tables is far from clear. For instance, in Table 1, $q_\theta$ has lower $-1$ and $-2.5$ scores, and I am not sure whether this can be referred to as "outperform". There might be a misunderstanding, but the authors should make it clear what we are expected to read from each table.

**Questions:**

No additional questions beyond the weaknesses above.

---

### Official Review · Reviewer_dU2W · 2024-11-04

**Soundness:** 2
**Presentation:** 1
**Contribution:** 3
**Rating:** 3
**Confidence:** 4

**Summary:**

This paper studies the problem of intractable inference in sequence models. It is proposed to sample MCMC chains using an infilling LM as a proposal, to create subchains using various procedures (e.g., selecting the points where the target energy decreases), and to train an autoregressive model on these subchains. This model can then be used to sample modes of the target distribution more efficiently. This method is evaluated on a protein design task and two language generation (editing) tasks and shows somewhat promising results.

**Strengths:**

- The procedure of amortising subchains of MCMC chains by a non-Markovian sequence model is interesting and could be applicable to structured sampling problems well beyond the protein and language tasks studied here.
  - For example, chain-of-thought reasoning, among other intractable inference problems in language, has been interpreted as latent variable inference and addressed using MCMC ([Phan et al.,  NeurIPS'23](https://arxiv.org/abs/2312.02179), [Lew et al.](https://arxiv.org/abs/2306.03081)), amortisation ([Hu et al., ICLR'24](https://arxiv.org/abs/2310.04363)), hybrid methods ([Zhao et al., ICML'24](https://arxiv.org/abs/2404.17546)), and distillation into tractable models ([Zhang et al., 2024](https://arxiv.org/abs/2406.13892)).
- The inclusion of both LM and biological sequence design tasks is a good way to show the generality (although toy experiments on something synthetic and *very* simple, where the target is well-understood, would also be helpful).
- Results generally comparable with prior work in each task with (possibly -- see questions below) higher efficiency of inference.

**Weaknesses:**

Overall, both the writing and the experiments need substantial improvement. It is quite difficult to understand the algorithm and the experiment setup, nor are the results particularly strong.

- The writing in the first two pages is imprecise and does not set up the problem well.
  - The first sentence of the abstract does not make sense to me. (Doesn't "desired outputs" already presuppose we are talking about generative modelling -- so what is the meaning of saying generative models are "appealing"? "Sequence-level" as opposed to what?)
    - In general, the abstract does not explain well the problem setting and does not prime the reader to expect what is to come. The reader is left wondering: Are we talking about generative modelling from data or sampling given a density? Are we training a new model or constraining an existing one?
  - The introduction does not set up the problem to be solved in a clear way.
    - The first 2.5 paragraphs or so of the introduction seem to be saying approximately "generalisation is important".
    - However, the **main objects** involved in the problem setting are not introduced. Thus it is not clear what is being talked about when "original" and "sampled" sequences are mentioned, what the EBM has to do with infilling, etc.
    - Second paragraph: Lu et al. and Holtzman et al. are inaccurately described. The latter studies strategies to clip the tail of the next-token distributions, while Lu et al. studies MCTS as proposed by reference [59] in that paper, which seems to have no connection to such clipping.
    - In the last paragraph of the intro, $q_\theta$ appears, but it is never defined.
- Section 2: Please define all symbols when they are first used. As far as I could infer:
  - $\cal X$: space of sequences
  - $\cal Y$: set of "properties"
  - $o:{\cal X}\to{\cal Y}$: oracle assigning a property to every sequence
  - ${\cal X}^{\rm train}$: training set
  - ${\cal Y}^{\rm train}$: unclear; is it $\{o(x):x\in{\cal X}^{\rm train}\}$? That is, we wish to generate sequences whose properties, as computed by an oracle, were not observed in the training set?
- Section 3:
  - Is the score giving the density or the log-density? Contradiction between "probability proportional to sequence-level score $s(x)$" (L144) and the equation in L149.
  - The following does not make sense to me (LL156-158): "masked language modeling objectives implicitly define EBM with the masked-infilling objective serving as a proposal distribution".
    - How can a [training] **objective** define a distribution? Do you mean that the trained model defines a distribution? Same in LL162-163: the self-supervised pre-training process seems to have nothing to do with the use of the trained infilling model (so how can we say that MH sampling is based on the self-supervised training?).
    - An EBM is defined by an energy function, not by a proposal distribution used to perform MH on it. Do you mean that the infilling model approximates a collapsed Gibbs sampler for a certain EBM? Or do you mean that the infilling model is an effective proposal for MH sampling of EBMs that happened to be trained on similar data?
    - Even in the former case, things are somewhat more subtle than this. The conditional distributions given by infilling may not be compatible with any joint distribution, see, for example, [Henningen and Kim, ACL'23](https://arxiv.org/abs/2305.15501) and [Wang and Cho, NeuralGen@NAACL'19](https://arxiv.org/abs/1902.04094).
    - In the end, I am left confused about the setting: the infilling model defines a EBM (L157) and is also a proposal distribution that is used to propose transitions (L160), but the proposals are accepted/rejected using the MH criterion on a EBM (L161), which seems to require an explicit energy model to compute the acceptance probability.
  - A little diagram would help to understand the different ways of forming trajectories for training $q_\theta$. (It is also confusing that $q$ (proposal) and $q_\theta$ (amortised sampler) are denoted by the same letter.)
- Experiments:
  - Missing details:
    - In each experiment, please specify exactly, in an equation, what the target energy model is. I found some notes on this in Appendix D, but they left me with more questions. In general, I would suggest to move some experiment details to the appendix, but state as directly as possible what sampling problem is being solved.
    - It is not stated which version of training data for $q_\theta$ is used. It can be inferred from the ablation study in Section 5, but would be good to make explicit.
  - I am not convinced by evaluations in 4.1:
    - The fluency is not reported for ICE.
    - Extrapolation metric: why should we expect that "make this negative" would lead necessarily to 1-star and not to 2-star reviews? It would be more informative to compare the full distributions of the base model and with editing, for each editing method.
      - How to understand that the second and third columns in Table 2 sum to more than 1, if they are the proportions of reviews with (in the example of negative sentiment) 2 and 1 stars, respectively?
      - In fact, generating **too many** 1-star reviews, together with the high fluency, could actually indicate mode collapse. There is no diversity metric to show that this is not happening.
  - The comment on diversity also applies to Section 4.2. Why was ICE not considered in this problem?
  - How do the methods compare in execution wall time (both for training and drawing a single sample)?

**Questions:**

Please see above.

---

### Official Review · Reviewer_iCrL · 2024-11-04

**Soundness:** 2
**Presentation:** 3
**Contribution:** 2
**Rating:** 5
**Confidence:** 4

**Summary:**

This paper proposes a method for learning sample-efficient extrapolative sequence transformations by first using Markov Chain Monte Carlo (MCMC) sampling to explore the solution space, then training a separate inference network on selected states from these chains. The authors demonstrate their approach on three tasks: protein sequence design, text sentiment control, and text anonymization. The key innovation is sub-sampling informative state transitions from MCMC chains to train a more efficient model that can achieve similar or better performance with significantly fewer inference steps.

**Strengths:**

- Tackles an important problem (extrapolative generation) with a novel approach
- Impressive empirical results, especially on protein engineering
- Thorough ablation studies examining different components
- Clear practical benefits in terms of sample efficiency

**Weaknesses:**

1. The validation methodology for protein engineering is fundamentally flawed. Without a separate validation set, there's no reliable way to assess generalization ability. While the authors acknowledge this and attempt some mitigation strategies, they don't address the core issue of potential overfitting to the test set through architecture and hyperparameter choices.

2. The theoretical justification for extrapolation capabilities is weak. Although they discuss how their approach captures dependencies between hidden states, they fail to provide a compelling explanation for why this enables better extrapolation. The paper lacks formal analysis of extrapolation capabilities or theoretical bounds on performance.

3. The baseline comparisons are inadequate, particularly in protein engineering where they only compare against two baselines from a single paper. This omits numerous recent protein design methods and alternative approaches. While sentiment and anonymization tasks include more baselines, they still miss obvious comparisons with state-of-the-art methods.

4. Several key ablation studies are missing. There's no exploration of different MCMC sampling strategies, limited analysis of energy function designs, and no investigation of how performance scales with sequence length. These missing analyses make it harder to understand which components are crucial for success.

5. The energy function design choices feel arbitrary and aren't well justified. The weighting between different terms lacks thorough explanation, and there's no meaningful discussion of how to design effective energy functions for new tasks, limiting the method's applicability to other domains.

6. Memory consumption issues are acknowledged but not adequately addressed. The authors note their models use significantly more memory than alternatives but offer no solutions. The analysis lacks discussion of how usage scales with sequence length or batch size, and fails to analyze memory-computation tradeoffs.

7. Reproducibility and practical concerns are significant. Many implementation details are buried in appendices, hyperparameter sensitivity isn't thoroughly analyzed, and there's limited discussion of failure cases or scalability to longer sequences. These omissions make it difficult for others to build upon their work or apply it to real-world problems.

**Questions:**

- Why not include more baselines, especially recent work in protein engineering?
- How sensitive is the method to the choice of energy function?
- Have you explored using curriculum learning approaches for training?
- How does performance scale with sequence length?
- What are the key factors limiting memory efficiency?

---

### Meta-Review · Area_Chair_c8LU · 2024-12-20

**Metareview:**

The paper proposes a method for controlled generation focused on extrapolating conditioning variables outside of the training distribution in the setting where a (possibly expensive) oracle scoring function exists. The work trains an AR model on chosen subsequence from MCMC chains with desired properties (such as increasing score). Experiments on protein sequence design, text sentiment control, and text anonymization and given. The authors claim their method achieves comparable to or better generalization than MCMC while being more sample efficient.

This is an important, and relevant problem. The method is interesting and while I've seen many works in the past which amortize MCMC though distillation into tractable models, this application is new and seems to be worth studying. The method can be applied to many domains and many types of target variable.

Reviewers believe that the method lacks theoretical justification for its extrapolation capabilities. When can is extrapolation like this possible? When is it now? The reviewers also found multiple issues with the paper's writing and experiments. As well they believed the method's limitations and failure cases were not discussed enough and many key experimental details were left out of the main paper.

While I find this to be an interesting problem and method, I will agree with the general consensus of the reviewers and recommend rejection.

**Additional Comments On Reviewer Discussion:**

The discussion period focused on a few key issues; the theoretical foundation of the approach, baseline comparisons, and the paper's presentation. They questioned if a method like this is capable of working in settings where traditional approaches cannot and felt that the paper left these important questions unanswered.

In response the authors added new baselines such as fudge but reviewers were not thoroughly convinced. They also had concerns about the validity of these new experiments for data access reasons. Authors clarified the setups should be comparable.

Still reviewers felt many of their largest questions were still unanswered and were not inclined to raise their scores.

---

### Decision · Program_Chairs · 2025-01-22

Reject